# Cuproptosis-related gene signatures define the immune microenvironment in diabetic nephropathy

**Hongmin Luo[1][�], Yuxuan Cao[2][�], Liping Guo[3], Hui Li[3], Yingying Yuan[3], Fan Lu[ID][3]\***

**1** Department of Nephrology, Hebei Medical University Third Hospital, Shijiazhuang, Hebei, China,
**2** Department of Epidemiology and Statistics, School of Public Health, Hebei Medical University, Hebei Key Laboratory of Environment and Human Health, Shijiazhuang, Hebei, China, **3** Department of Nephrology, The Fourth Hospital of Hebei Medical University, Shijiazhuang, Hebei, China

९ These authors contributed equally to this work and were co-first authors.
\* lufande@126.com

## Abstract

### Background

Cuproptosis may be a new clue to illustrate the pathogenesis of the disease. There was no study focused on the relationship between the cuproptosis genes and diabetic nephropathy (DN). This study aimed to reveal the relationship between cuproptosis genes and the immune microenvironment in DN and distinguish different phenotypes to describe disease heterogeneity through consensus clustering based on cuproptosis genes.

### Methods

We downloaded RNA sequencing data sets of DN glomerular and normal renal tissue samples (GSE142025, GSE30528, and GSE96804) from the Gene Expression Omnibus (GEO) database. Differentially expressed genes (DEGs) between DN and control samples were screened. Immune cell subtype infiltration and immune score were figured out via different algorithms. Consensus clustering was performed by Ward's method to determine different phenotypes of DN. Key genes between phenotypes were identified via a machine-learning algorithm. Logistic regression analysis was applied to establish a nomogram for assessing the disease risk of DN. The role of related genes was verified by cell experiments.

### Results

In DN samples, NOD-like receptor thermal protein domain associated protein 3(NLRP3) and cyclin-dependent kinase inhibitor 2A Gene(CDKN2A) were positively correlated to immune score. Nuclear factor erythroid 2-related factor 2(NFE2L2), Lipoic Acid Synthetase(LIAS), Lipoyltransferase 1(LIPT1), Dihydrolipoamide

**Data availability statement:** All relevant data are within the paper and Supporting Information files.

**Funding:** This work were supported by Hebei Medical Science Research Key Project, received by Liping Guo, Hui Li, Yingying Yuan and Fan Lu (No. 20221236) and Hebei Natural Science Foundation, received by Hongmin Luo and Yuxuan Cao (No. H2023206295).

**Competing interests:** No authors have competing interests.

dehydrogenase(DLD), Dihydrolipoamide Branched Chain Transacylase E2(DBT) and Dihydrolipoamide S-Succinyltransferase(DLST) were negatively correlated to immune score. Via Consensus clustering based on cuproptosis genes, the DN samples were divided into cluster C1 and cluster C2. Cluster C1 was characterized by low cuproptosis gene expression, high immune cell subtype infiltration, and high enrichment of immune-related pathways. Cluster C2 was on the contrary. Dicarbonyl/l-xylulose reductase (DCXR) and heat-responsive protein 12 (HRSP12) were key genes related to clinical traits and immune microenvironment, negatively correlated with most immune cell subtypes. The nomogram constructed based on DCXR and HRSP12 showed good efficiency for DN diagnosis.

## Conclusion

Immune microenvironment imbalance and metabolic disorders may lead to the occurrence of DN. Cuproptosis genes, with the ability to regulate the immune microenvironment and metabolism, can be used for disease clustering to describe the heterogeneity and characterize the immune microenvironment. HRSP12 and DCXR, as key genes related to disease phenotypes and immune microenvironment characteristics, were jointly constructed as nomograms for DN diagnosis with high accuracy and reliability. HRSP12 and DCXR may be potential biological markers and renal protective factors.

## Introduction

Diabetic nephropathy (DN), as a common complication of diabetes mellitus (DM), may lead to end-stage renal disease [1]. The incidence of DM is increasing year by year, which may be related to changes in dietary structure and lifestyle habits caused by social development [2,3]. The incidence of diabetic complications, including DN, also is increased [4,5]. DN patients accounted for about 1/3 to 1/2 of the total number of DM patients [6]. Once diabetic nephropathy occurs, renal function declines sharply and rapidly progresses to end-stage renal disease without timely medical treatment [7,8]. Although now some treatments can relieve the clinical symptoms caused by DN and delay the progression of the disease [9], patients still have to face poor prognosis, heavy financial burden and great psychological pressure [10].

In view of the increasing incidence of DN, the clinical features of rapid progression and poor prognosis, the lack of early screening indicators and effective therapeutic means, it is significant to elucidate the pathogenesis of DN, and to search for potential biological markers and new therapeutic targets [11].

Cuproptosis, as a new cell death mode, is associated with the occurrence and progression of many diseases [12–15]. Cuproptosis genes may regulate the metabolism and immune microenvironment [16–18]. The metabolic disorder and immune response induced renal fibrosis are considered to be the main pathological mechanisms of DN [19]. Cuproptosis may be a new clue to illustrate the pathogenesis of the

disease. There was no study focused on the relationship between the cuproptosis genes and DN. Considering the regulatory effects of cuproptosis genes on immune microenvironment and substance metabolism, this study was aimed to reveal the relationship between cuproptosis genes and immune microenvironment in DN and distinguish different phenotypes to describe disease heterogeneity through consensus clustering based on cuproptosis genes.

In this study, we explored the correlation between cuproptosis genes and immune cell subtypes, divided DN samples into clusters with different immune microenvironment characteristics and different metabolic characteristics via consensus clustering, screened key genes related to different phenotypes, constructed disease diagnosis models, and revealed the regulatory effects of key genes on immune microenvironment. This study was expected to provide new ideas and clues for the early diagnosis and targeted therapy of DN.

## Methods

### Data download

The detailed flow chart of the research process is shown in Fig 1.

We downloaded RNA sequencing data sets of DN glomerular tissue samples and normal renal tissue samples (GSE142025, GSE30528, and GSE96804) from Gene Expression Omnibus (GEO) database. GSE142025 consisted of 28 DN glomerular tissue samples and 9 control tissue samples. GSE30528 consisted of 9 DN glomerular samples and 13 control tissue samples. GSE96804 consisted of 41 DN glomerular tissue samples and 20 control tissue samples. We performed batch normalize on three data sets. We combined GSE30528 and GSE96084 into one data matrix as the training cohort and GSE142025 as the test cohort.

### Differentially expressed genes (DEGs) and enrichment analysis

We used R software and R package (limma) to screen DEGs. Fold change = DN samples gene expression/ control samples gene expression. Genes with |logFC|>0.5 and false discovery rate (FDR)<0.05 were identified as DEGs [20]. The results were visualized as volcano plot and heatmap using the "ggplot2" and "pheatmap" R packages. Subsequently, functional enrichment analysis of DEGs, including Gene Ontology (GO) and Kyoto Encyclopedia of Genes and Genomes (KEGG), were performed using the "clusterProfiler" R package [21]. The pathways with $P<0.05$ were identified as significant.

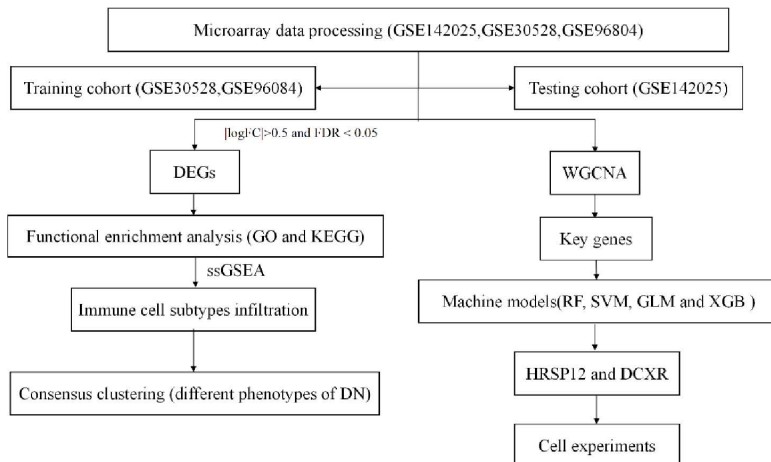

**Fig 1. Study flow chart.**

## Evaluation of immune mircoenvironment

Immune cell subtypes infiltration was figured out by ssGSEA algorithm. Immune score, stromal score and ESTIMATE score were figured out by ESTIMATE algorithm [22].

## Consensus clustering

Consensus clustering was performed to determine different phenotypes of DN according to the expression profiles of cuproptosis genes. The cluster numbers and robustness were assessed by consensus clustering. We ran 1000 iterations of the above steps to guarantee the classification robustness with the R package "ConsensuClusterPlus" [23]. We identified the value of K that flattened the cumulative distribution function curve within the range of 0.1 to 0.9 of the consensus index.

## Weighted gene co-expression network analysis (WGCNA)

We used WGCNA to construct gene coexpression networks and explored modules highly associated with DN [24]. The correlation between modules and clinical features was calculated. The cut height was set to 0.25 and each module contained at least 30 genes. The correlation between modules and phenotypes, gene significance, and module membership were calculated.

## Construction of predictive model based on multiple machine learning methods

Based on two different cuproptosis genes clusters, we applied the "caret"R packages (version 6.0.91) for establishing machine learning models including random forest model (RF), support vector machine model (SVM), generalized linear model (GLM), and eXtreme Gradient Boosting (XGB) [25]. The distinct clusters were considered as the response variable, and the cluster-specific DEGs were selected as explanatory variables. The 83 DN samples were randomly classified into a training set (60%, N = 50) and a validation set (40%, N = 33). The caret package automatically tuned the parameters in these models by grid search, and all of these machine learning models were performed with default parameters and assessed via 5-fold cross validation. The "DALEX" package (version 2.4.0) was carried out to interpret the aforementioned four machine learning models and visualize the residual distribution and feature importance among these machine learning models. The "pROC" R package (version 1.18.0) was performed to visualize the area under ROC curves. Consequently, the optimal machine learning model was determined and the top two important variables were considered as the key predictive genes associated with DN. Finally, The ROC curves analysis were performed to verify the diagnostic value of the diagnostic model.

## Cell culture, treatment and transfection

Human kidney-2 (HK-2) cells were cultured with DMEM medium(Zeta Life, USA) containing 10% FBS(Zeta Life, USA), 100 U/mL penicillin, and 100 µg/mL streptomycin placed in a 5% $CO_2$ incubator at 37 °C. The cells were treated with 30 mmol/l glucose(Solarbio, China) for 24h.

ShRNA lentiviral vectors targeting HRSP12 and DCXR were purchased from Shanghai Jikai Gene Technology Co., Ltd. The bacterial solution was shaken in LB culture medium containing ampicillin at 220 rpm and incubated at 37 °C for 15h. Plasmid assay concentrations were collected using the EndoFree Mini Plasmid Kit (TIANGEN, China) and quality checked. HK-2 cells treated with high glucose were cultured adherently at a density of $1X10^5$ in a 6-well plate, and replaced with serum-free culture medium when the cell fusion reached 50%~60%. Prepared working solution: opti-MEM 300 µL, E-Trans DNA transfection reagent 5 µL and plasmid 2.5 µg, mixed well, placed at room temperature for 10min, added to a 6-well plate to transfect cells, cultured in a cell culture incubator, and observed the transfection of cells at each time point (24h and 48h) under a fluorescence microscope.

## Quantitative real-time PCR analysis

We used TRIpure Reagent (Aidlab, China) to extract total RNA from cells, reversed transcribe it with FastKing cDNA first-strand synthesis kit(TIANGEN, China) and used SuperReal enhanced version of the fluorometric premixed reagent (TIANGEN, China) for quantitative real-time PCR. The ΔΔCT method was used to determine the relative changes in gene expression normalized to cells GAPDH gene expression.

## Cell Counting Kit-8 (CCK-8) assay

HRSP12-shRNA cells, DCXR-shRNA cells, and shNC cells in the logarithmic growth phase were digested and centri-fuged, and 2 ml of fresh medium was added to resuspend. Count under the microscope. 2000 cells/200 μL of cell suspen-sion per well were added to the 96-well plate, the edge wells were filled with sterile PBS 200 μL, and 5 double wells were set up in each group and cultured at 37°C and 5% $CO_2$ conditions. During the detection, the fresh medium was replaced and CCK-8 solution 10μL/well was added, and the absorbance value of each well was detected at 450nm wavelength at 37°C and 5% $CO_2$ for 2h. A total of 6 days were detected in each group of cells. The cell growth curve was plotted with the time abscissa and the absorbance value as the ordinate.

## Statistical analysis

Logistic regression analysis was applied to establish an nomogram for assessing disease risk. Correlation analysis was conducted via Spearman test. Receiver operator characteristic curve (ROC) was conducted to evaluate diagnostic efficacy. *P*-value <0.05 on both sides indicated statistical significance. The data sets obtained were open source with no require of ethics committee review. All data in this study are open source and do not require ethics committee review.

# Results

## DEGs screening and enrichment analysis

Compared with the control samples, 548 genes were down-regulated and 364 genes were up-regulated in the DN sam-ples (Fig 2A and 2B, S1 Table). The up-regulated genes in DN samples were enriched in biological process items, such as inflammatory response, and cell adhesion. The up-regulated genes in DN samples were enriched in cell component items, such as collagen-containing extracellular matrix. The up-regulated genes in DN samples were enriched in molec-ular function items, such as extracellular matrix structural constituent (Fig 2C, S2 Table). The up-regulated genes in DN samples were enriched in pathway, such as PI3K-Akt signaling pathway and cytokine-cytokine receptor interaction, which were related to immune response (Fig 2D, S3 Table). The down-regulated genes in DN samples were enriched in bio-logical process items, such as glomerulus development. The down-regulated genes in DN samples were enriched in cell component items, such as extracellular exosome and cytosol. The down-regulated genes in DN samples were enriched in molecular function items, such as protein binding (Fig 2E, S4 Table). The down-regulated genes in DN samples were enriched in pathway, such as metabolic pathwaysand arginine and proline metabolism (Fig 2F, S5 Table).

## Immune microenvironment of DN

We calculated the relative abundance of 28 immune cell subtypes in DN and control samples. Activated B cell, natural killer T cell, natural killer cell and other immune cell subtypes were enriched in DN samples, while CD56 bright natural killer cell, immature dendritic cell, Type 17 T helper cell and other immune cell subtypes were enriched in control samples (*P*<0.05, Fig 3A). Stromal score, immune score and ESTIMATE score of DN samples were higher than those of control samples (*P*<0.05, Fig 3B). In DN samples, the abundance of most immune cell subtypes were positively correlated, but immature dendritic cell and Type 17 T helper cell were negatively correlated with that of other immune cell subtypes (*P*<0.05, Fig 3C). NLRP3 and CDKN2A as cuproptosis genes were positively correlated to immune score. NFE2L2, LIAS,

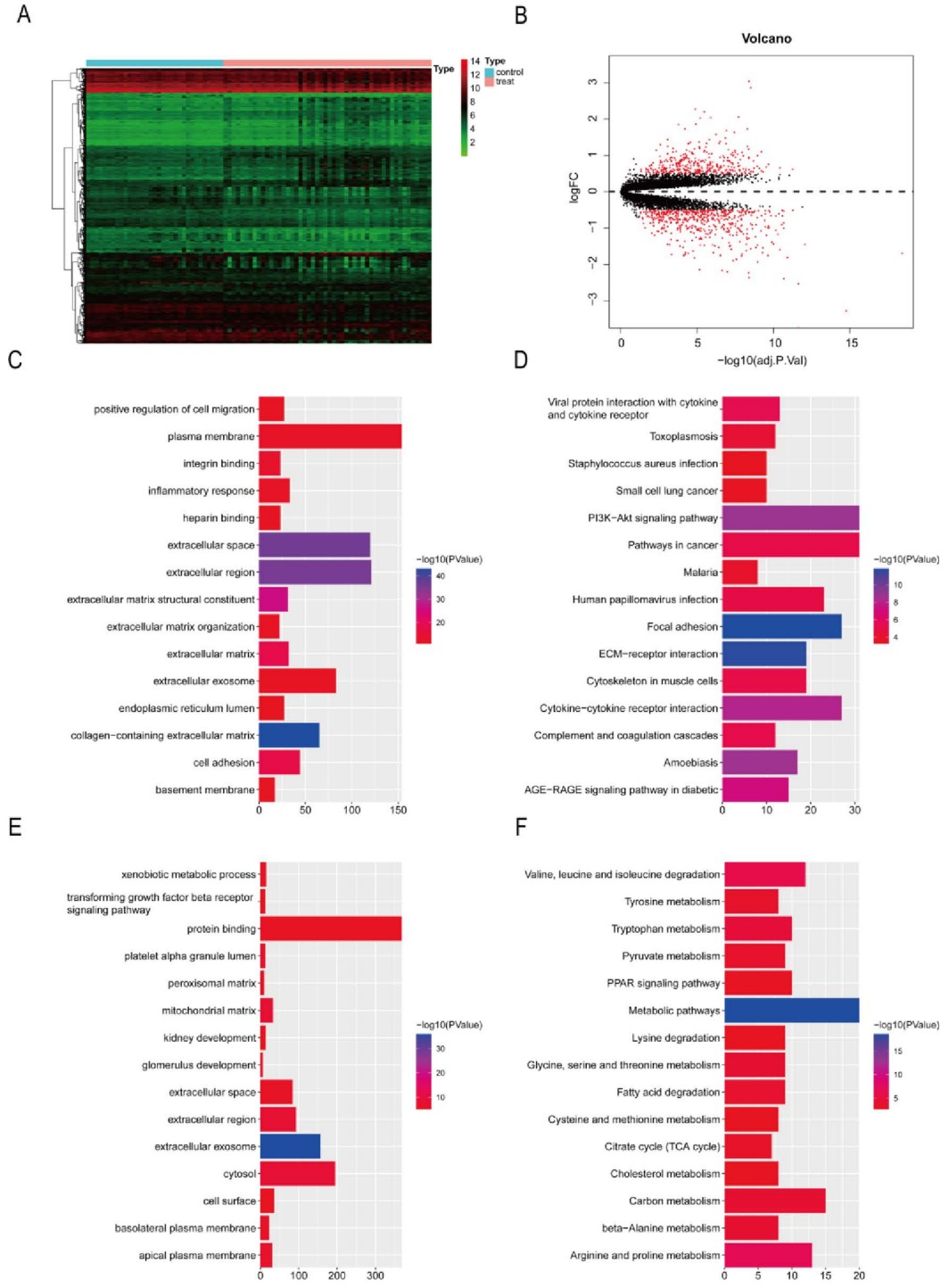

**Fig 2. Identification of differentially expressed genes (DEGs).** (A) The heatmap of DEGs between control samples and diabetic nephropathy (DN) samples. The abscissa represents the clustered samples, and the ordinate represents DEGs. Red indicates up-regulated expression, while blue indicates down-regulated expression. (B) Volcano of DEGs. (C) Gene Ontology (GO) enrichment analysis of up-regulated genes in DN samples. (D) Kyoto Encyclopedia of Genes and Genomes (KEGG) enrichment analysis of up-regulated genes in DN samples. (E) Gene Ontology (GO) enrichment analysis of down-regulated genes in DN samples. (F) KEGG enrichment analysis of down-regulated genes in DN samples.

LIPT1, DLD, DBT and DLST as cuproptosis genes were negatively correlated to immune score ($P < 0.05$, Fig 3D). NLRP3 and CDKN2A were positively correlated to the most immune cell subtypes, while NFE2L2, LIAS, LIPT1, and DBT were negatively correlated to the most immune cell subtypes ($P < 0.05$, Fig 3E).

## Consensus clustering based on cuproptosis genes

We compared the differences in the expression of cuproptosis genes between the control samples and the DN samples. Compared with the control samples FDX1, LIAS and DLAT were highly expressed in DN samples, while DBT was low expressed in DN samples ($P < 0.05$, Fig 4A). Based on the differentially expressed cuproptosis genes, consensus cluster-ing was performed. When K = 2, the cumulative distribution function curve is relatively flat within the range of consensus index from 0.1 to 0.9 (Fig 4B). The DN samples were divided into cluster C1 and cluster C2, including 34 and 17 samples respectively (Fig 4C). PCA showed that in the plane dimension composed of principal component 1 and principal com-ponent 2, the two clusters of samples were enriched in different regions (Fig 4D). Cluster C1 was characterized by low expression of cuproptosis genes. On the contrary, the cuproptosis genes were highly expressed in cluster C2 ($P < 0.05$, Fig 4E and 4F). GSVA revealed that immune-related pathways were more significant in the cluster C1. Metabolism-related pathways were more significant in the cluster C2 (Fig 4G, S6 Table). Compared with cluster C2, gamma delta T cell, immature B cell, natural killer T cell, natural killer cell, plasmacytoid dendritic cell, T follicular helper cell and cen-tral memory CD4 T cell were more enriched in the samples of cluster C1 ($P < 0.05$, Fig 3H). Compared with cluster C2, CD56dim natural killer cell, immature dendritic cell and type 17 T helper cell were less enriched in the samples of cluster C1 ($P < 0.05$, Fig 4H).

## Identification of key genes based on WGCNA

We first constructed a gene-weighted coexpression network between cluster C1 and cluster C2. The MEDissThres were set to 0.25 to combine modules with similar distances. A total of five modules were obtained (Fig 5A). The correla-tion between different modules and clusters was calculated. We reserved modules with $P < 0.05$, and turquoise module (|cor| = 0.73, $P = 1e-9$) and brown module (|cor| = 0.3, $P = 0.03$) were reserved (Fig 5B). We continued to screen out key genes in the module, and the screening criteria were gene significance > 0.5 and module membership > 0.8. While 240 genes in the turquoise module met the screening criteria, none in the brown module did.

Secondly, we constructed a gene-weighted co-expression network between control samples and DN samples. The MEDissThres were set to 0.25 to combine modules with similar distances. A total of six modules were obtained (Fig 5C). The correlation between different modules and clinical features was calculated. We reserved modules with $P < 0.05$, and blue module (|cor| = 0.39, $P = 3e-4$), brown module(|cor| = 0.55, $P = 1e-7$), green module (|cor| = 0.28, $P = 0.01$), turquoise module (|cor| = 0.5, $P = 2e-6$) and yellow module (|cor| = 0.58, $P = 8e-09$) were reserved (Fig 4D). We continued to screen out key genes in the module, and the screening criteria were gene significance greater than 0.5 and module membership exceeding 0.8. While 127 genes in the reserved modules met the screening criteria.

The DEG set, the WGCNA (cluster C1 vs Cluster C2) gene set and the WGCNA (control samples vs DN samples) gene set had eight overlapping genes, CYB5A, DCXR, GRHPR, HRSP12, MAOA, SDHB, SLC9A3R1, TST (Fig 5E).

## Construction and assessment of machine learning models

To further identify subtype-specific genes with high diagnostic value, we established four proven machine learning models [random forest model (RF), support vector machine model (SVM), generalized linear model (GLM), and eXtreme Gradient Boosting (XGB)] based on the expression profiles of 8 cluster-specific DEGs in the DN training cohort. The "DALEX" package was applied for explaining the four models and plotting the residual distribution of each model in the test set. RF, GLM and SVM machine learning models presented a relatively lower residual (Fig 6A). Subsequently, the

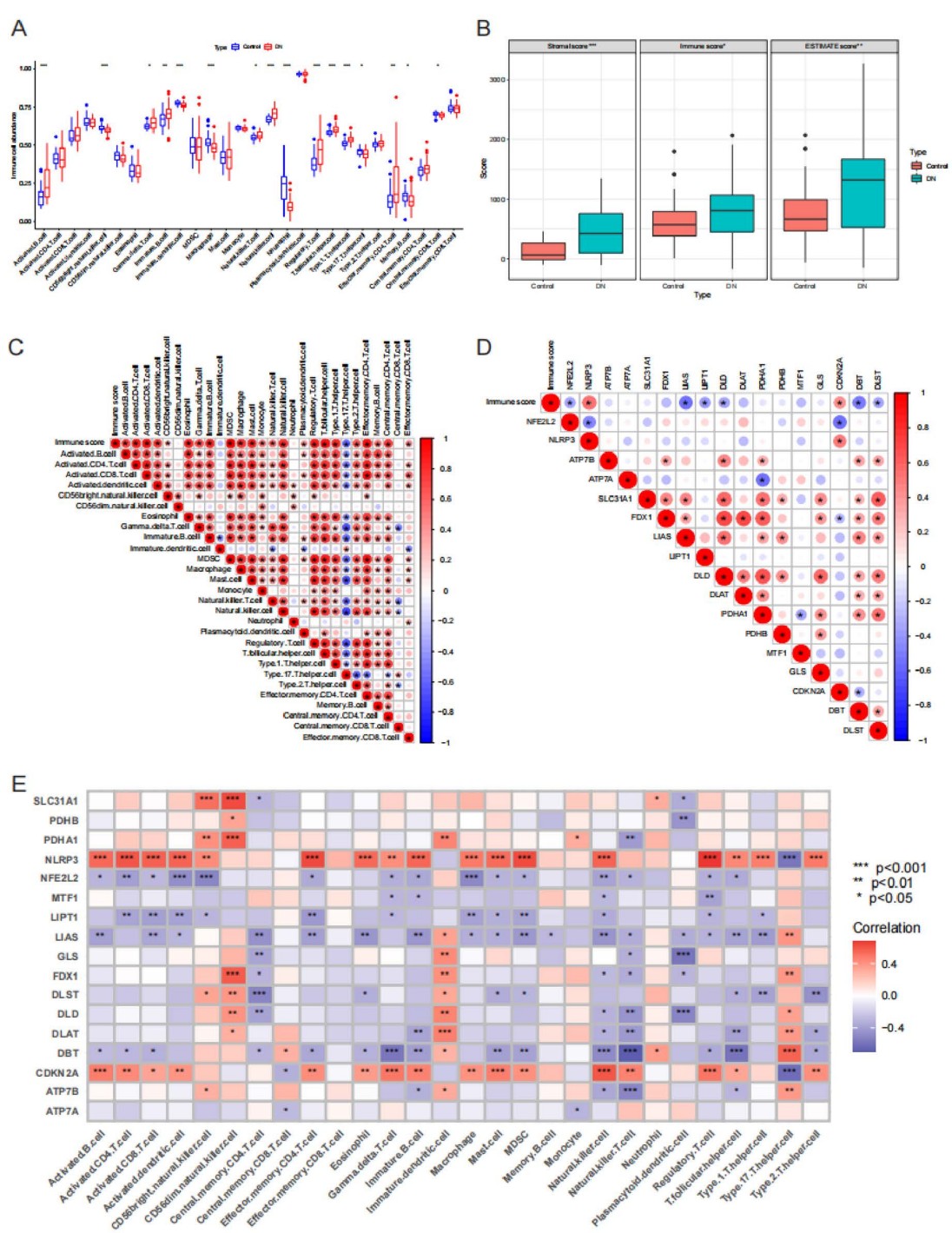

**Fig 3. Immune microenvironment of diabetic nephropathy (DN) samples.** (A) Differences of immune cell subtypes between control samples and DN samples. (B) Differences of stromal score, immune score and ESTIMATE score between control samples and DN samples. (C) The correlation between immune score and immune cell subtypes. Different colors indicate different correlation coefficients. (D) The correlation between immune score and cuproptosis genes. Different colors indicate different correlation coefficients. (E) The correlation between immune cell subtypes and cuproptosis genes. * represents P < 0.05, ** represents P < 0.01, *** represents P < 0.001.

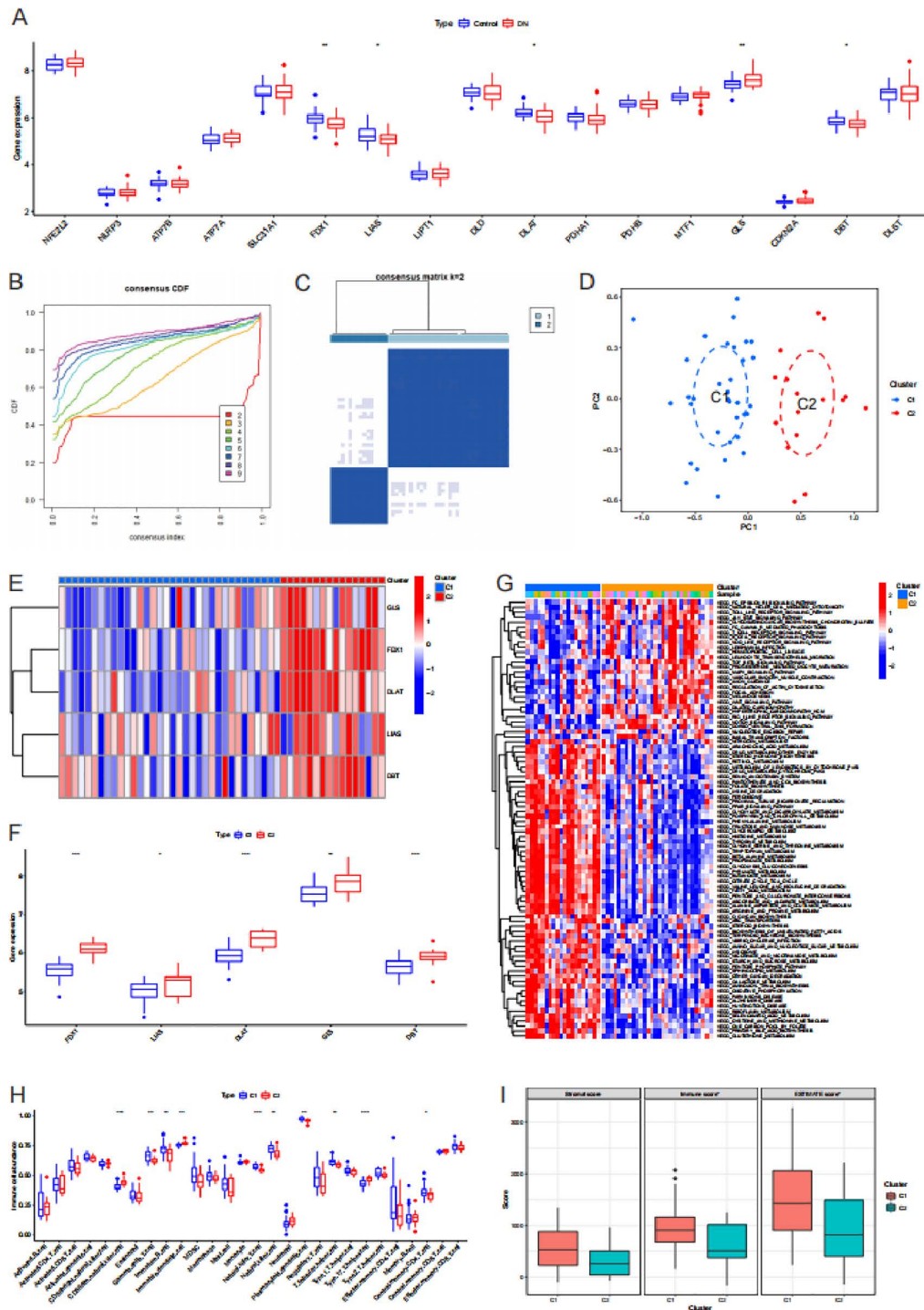

**Fig 4. Consensus clustering base on expression of cuproptosis genes.** (A) Differences of cuproptosis genes between control samples and diabetic nephropathy (DN) samples. (B) The cumulative distribution function (CDF) curves for different values of k. Different colors represent different K values. (C) Consensus clustering matrix with K=2. (D) Principal component analysis of DN samples based on gene expression. (E) Heatmap of cuproptosis genes between cluster C1 and cluster C2. (F) Differences of cuproptosis genes between cluster C1 and cluster C2. (G) Gene set variation analysis of cluster C1 and cluster C2. (H) Differences of immune cell subtypes between cluster C1 and cluster C2. B Differences of stromal score, immune score and ESTIMATE score between cluster C1 and cluster C2.

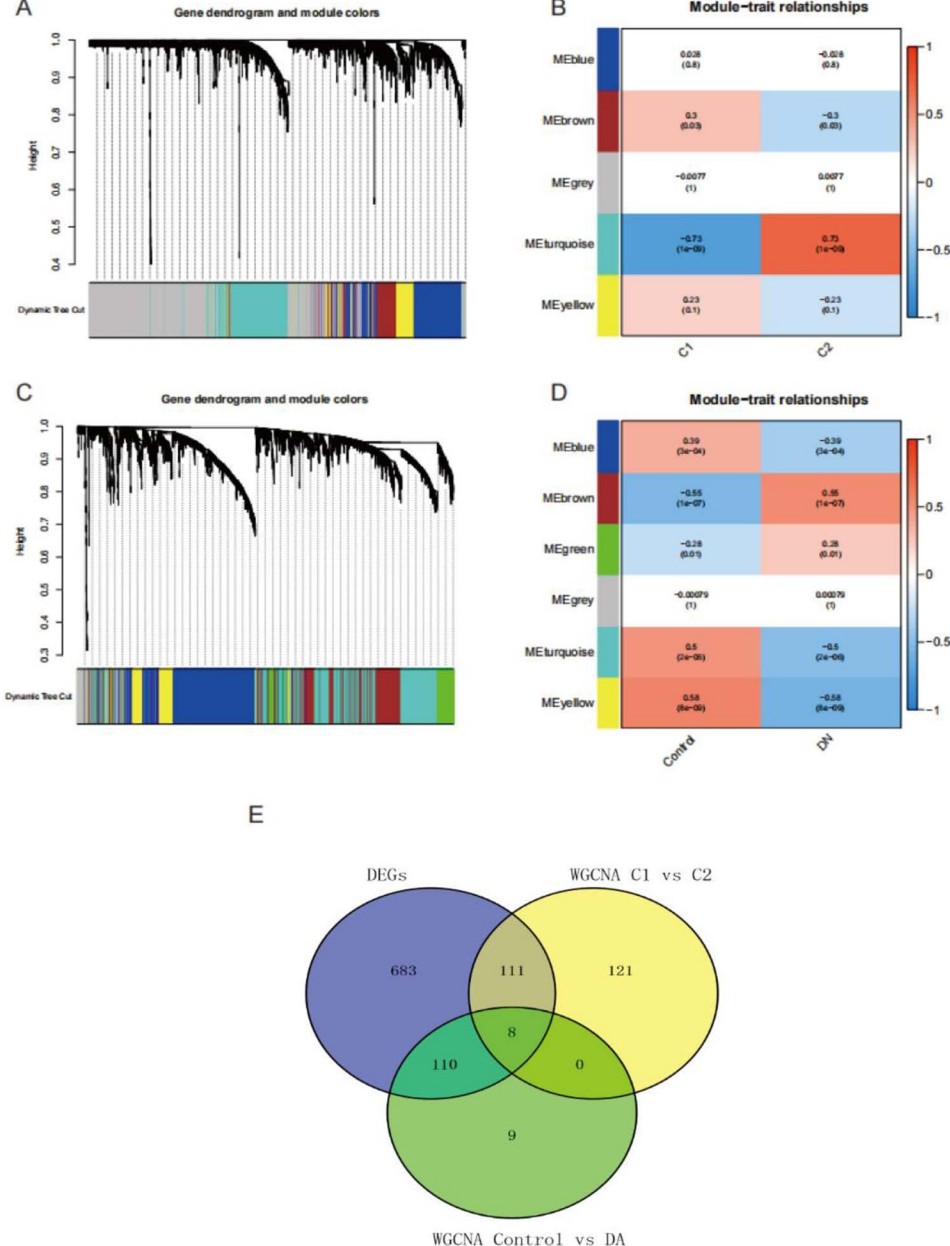

**Fig 5. Identification of key genes of different phenotypes.** (A) Dynamic tree cut to get modules related to cluster C1 and cluster C2. (B) The correlation between modules and clusters. Different colors indicate different correlation coefficients. The value in parentheses represents P-value. (C) Dynamic tree cut to get modules related to control and DN samples. (D) The correlation between modules and control and DN sample sets. Different colors indicate different correlation coefficients. The value in parentheses represents P-value. (E) Intersection of differentially expressed genes and phenotypic key genes.

top 2 important feature variables of each model were ranked according to the root mean square error (RMSE) (S7 Table). Moreover, we evaluated the discriminative performance of the four machine learning algorithms in the testing set by calculating receiver operating characteristic (ROC) curves based on 5-fold cross-validation. The SVM machine learning model displayed the highest area under the ROC curve (AUC) (SVM, AUC = 0.881; RF, AUC = 0.854; XGB, AUC = 0.792;

A

B

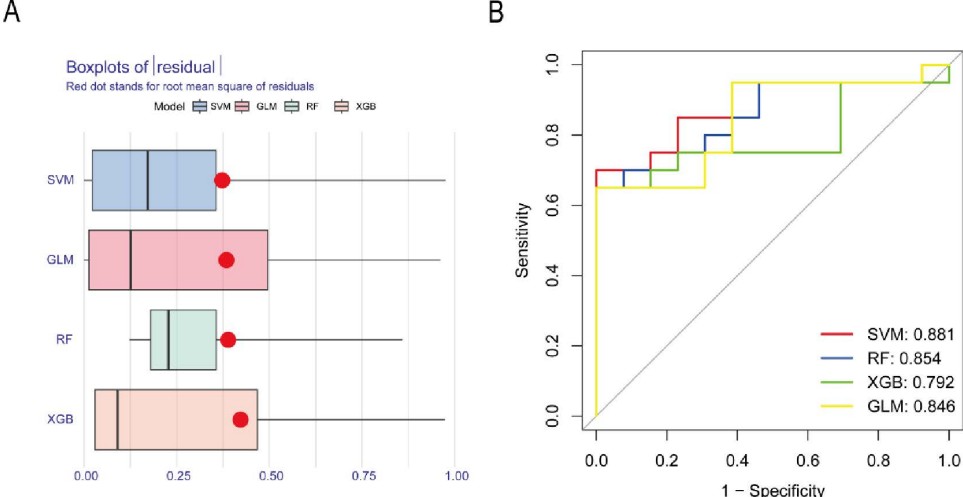

**Fig 6. Construction and evaluation of RF, SVM, GLM, and XGB machine models.** (A) Boxplots showed the residuals of each machine learning model. Red dot represented the root mean square of residuals (RMSE). (B) ROC analysis of four machine learning models based on 5-fold cross-validation in the testing cohort.

GLM, AUC = 0.846, Fig 6B). Overall, combined with these results, the SVM model was demonstrated to best distinguish patients with different clusters. Finally, the top five most important variables (HRSP12 and DCXR) from the SVM model were selected as predictor genes for further analysis.

### Nomogram construction and evaluation

Based on the expression of DCXR and HRSP12, we constructed a nomogram model for DN diagnosis (Fig 7A). The calibration curve showed that the bias-corrected curve fitted the ideal curve (n = 83, mean absolute error = 0.034, mean squared error = 0.0016, quantile of absolute error = 0.061) (Fig 7B). The decision tree curve showed that the model curve deviated from all curve, and the model had good accuracy (Fig 7C). ROC curve showed that AUC of training cohort and test cohort was 0.874 and 0.822 respectively (Fig 7D and 7E).

### The relationship between key genes and immune microenvironment

Compared with cluster C1, DCXR and HRSP12 were highly expressed in the samples of cluster C2 (P < 0.001, Fig 8A and 8B). Compared with the control samples, DCXR and HRSP12 were lowly expressed in DN samples(P < 0.001, Fig 8C and 8D). In DN samples, DCXR (r = -0.45, P = 0.0013) and HRSP12 (r = -0.53, P = 8.8e-5) were negatively correlated with immune score (Fig 8E and 8F). DCXR and HRSP12 were negatively correlated with most immune cell subtypes (Fig 8G).

### Effect of DCXR and HRSP12 on DN cells

We compared the expression of DCXR and HRSP12 in normal HK-2 cells and in DN model HK-2 cells and found that they were higher in DN model(P < 0.05, Fig 9A). After that, we performed CCK8 experiments on HK-2 cells in the DN model by knocking out DCXR and HRSP12, and found that compared with the negative control, the cell proliferation increased after knocking out the genes(P < 0.0001, Fig 9B), and the results showed that the DCXR and HRSP12 genes played a role in inhibiting cell proliferation in the DN model, and the DCXR and HRSP12 genes were the protective factors of the kidney.

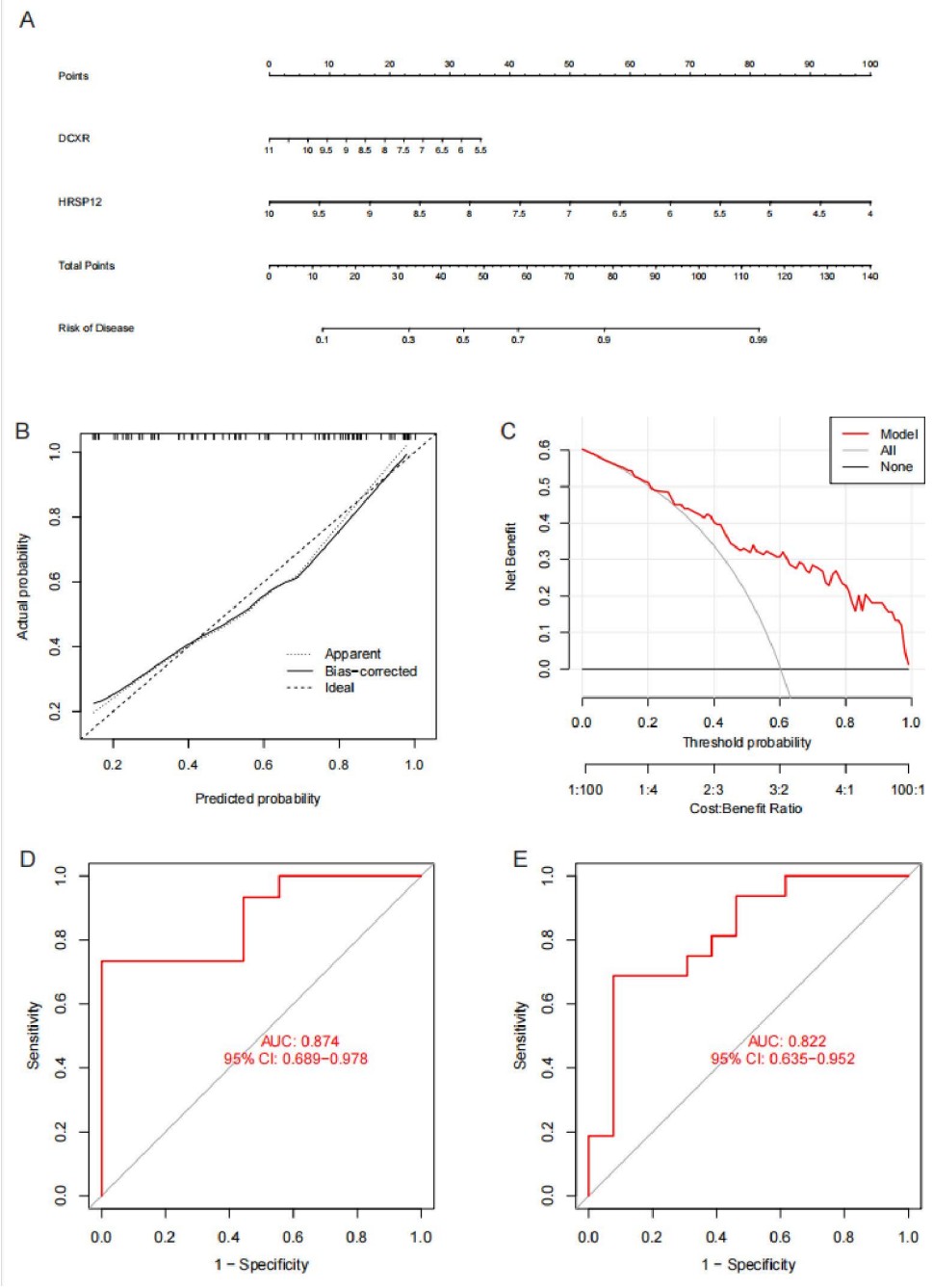

**Fig 7. Nomogram construction and evaluation.** (A) Nomogram for risk of disease. (B) Calibration curve for nomogram. (C) Decision-making tree curve for nomogram. (D) Receiver operating characteristic (ROC) curve for nomogram of training cohort. (E) ROC curve for nomogram of test cohort.

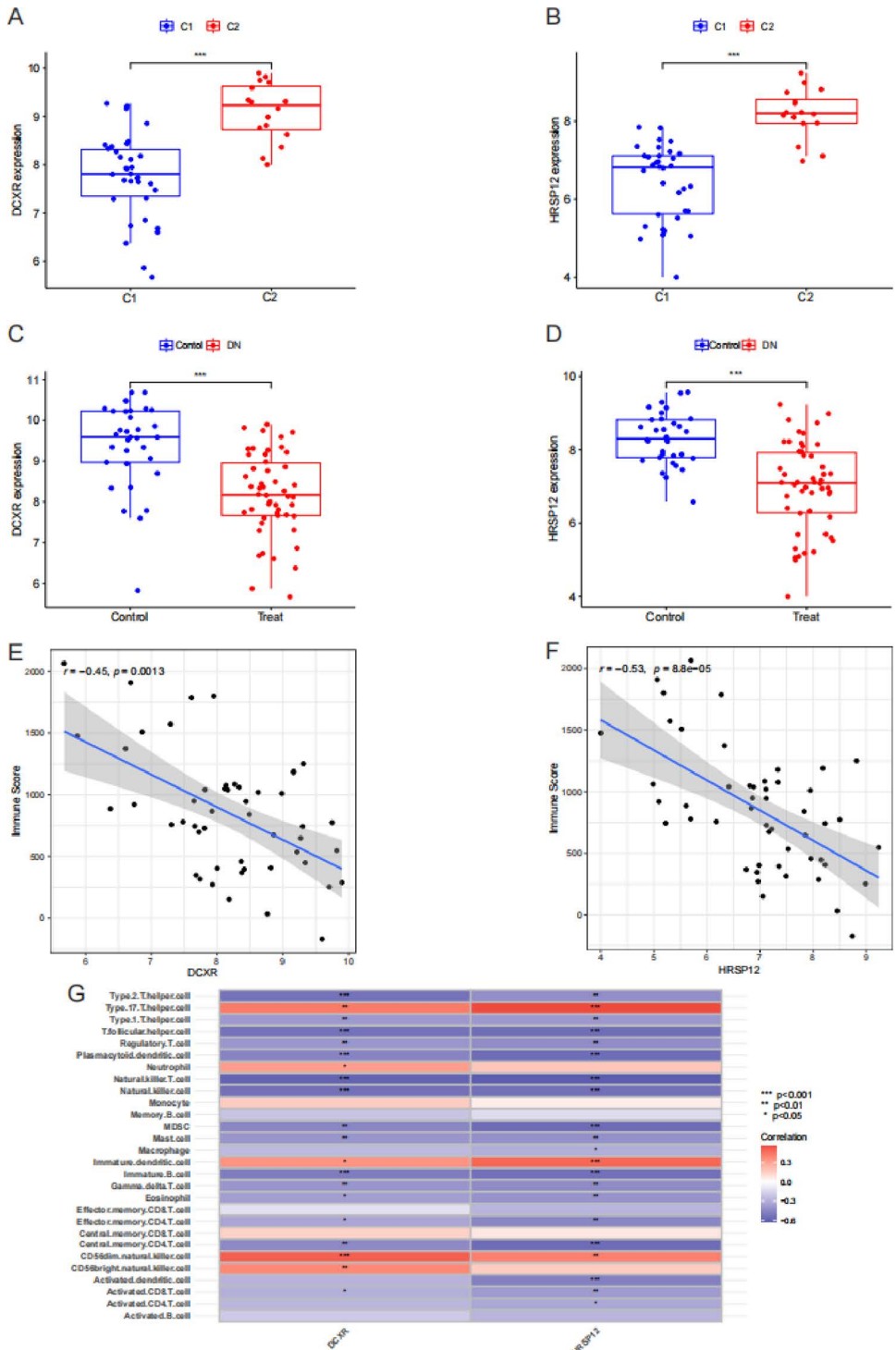

**Fig 8. The relationship between key genes and immune microenvironment.** (A and B) DCXR and HRSP12 expression between control samples and DN samples. (C and D) DCXR and HRSP12 expression between cluster C1 and cluster C2. (E and F) The correlation between DCXR and HRSP12 and immune score. (G) The correlation between DCXR and HRSP12 and immune cell subtypes.

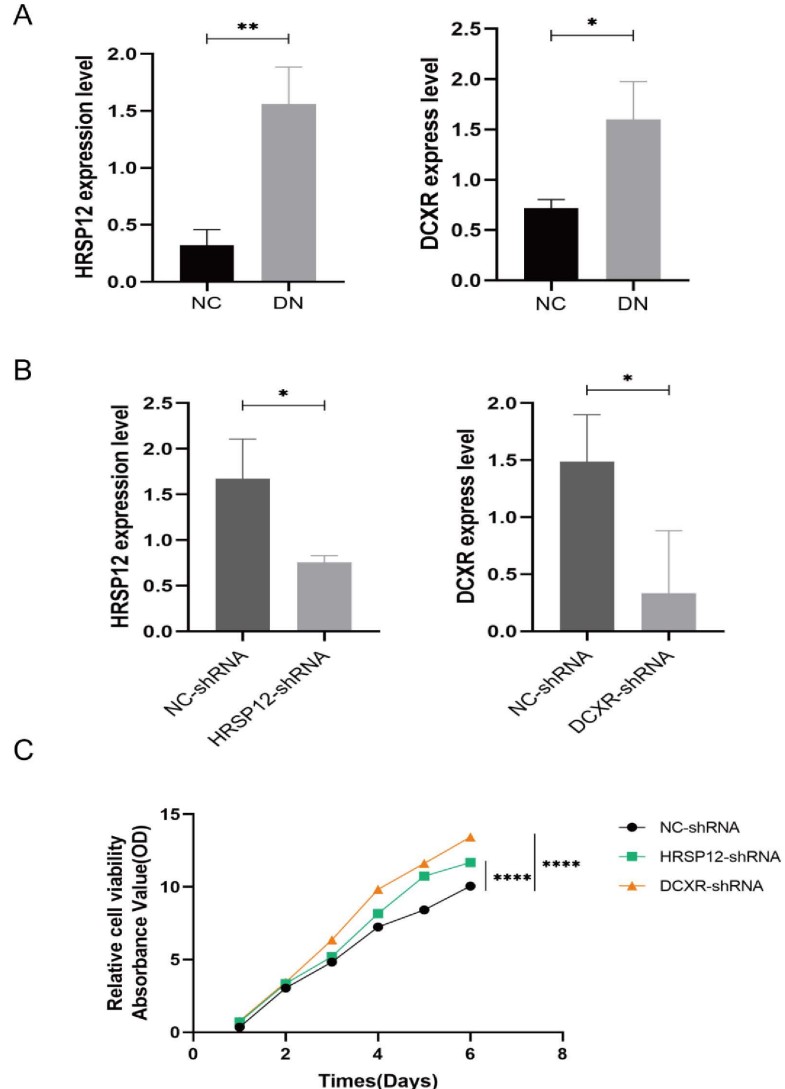

**Fig 9. (A) The expression of DCXR and HRSP12 was higher in the DN model; (B) The expression of DCXR and HRSP12 was decreased compared with the normal control after knockout; (C) CCK8 experiment.**

## Discussion

Cuproptosis is related to the occurrence and development of diseases. The main pathological mechanism of DN is metabolic disorder and renal fibrosis induced by immune response. This study is the first to link Cuproptosis gene with DN, and explore the possibility of Cuproptosis affecting the pathogenesis of DN. The relationship between Cuproptosis gene and immune microenvironment in DN was revealed, and the heterogeneity of disease was explored. Two key genes, HRSP12 and DCXR, were not confirmed in previous studies to have a protective effect of DCXR on kidney. This study verified this conclusion from the results of cell experiments.

We first screened out DEGs in normal glomerular tissue and DN glomerular tissue. The upregulated genes in DN samples were significantly enriched in immune-related functional items and signaling pathways. These results suggested that the immune response may lead to the occurrence of DN. This was consistent with the results of some previous studies.

According to the study of Lampropoulou et al., in the early stage of DN disease, TNF-α signaling pathway and T cells activation began to play a pathogenic role, showing a positive correlation between the two, which was believed to have a coordination role. T cell markers and inflammatory cytokines were elevated as the disease progresses [26]. According to the study of Lemos et al., the main pathological mechanism of kidney injury in DN was that inflammation promoted renal tissue fibrosis. IL-1β was an important link between inflammation and renal fibrosis [27]. A variety of chemokines, such as IL-6, TNF-α and IL-18, as well as a variety of inflammatory response-related mechanisms, such as OS signaling pathways, PI3K/AKT/mTOR signaling pathway, were regard as pathogenic factors in the process of renal injury in DN [6,28–34].

We further explored the immune microenvironment in both clusters of samples. We found that the immune score of DN samples was higher than that of normal samples. Most immune cell subtypes were enriched in DN samples. In DN samples, there was a trend of positive correlation between immune cell subtypes. These results suggested that changes in the immune microenvironment may be one of the essential features of the disease. In the course of disease occurrence and progression, immune cells continued to differentiate and accumulated in the diseased tissues. Immune response may be an important pathogenic mechanism. Several studies explored changes in the immune microenvironment of DN. Previous studies pointed out that the higher macrophages infiltration in kidney tissue, the worse the renal function of patients, the worse the prognosis, which were thought to be a driver of renal fibrosis [35–38]. Inhibition of T cell proliferation and dendritic cell activation can protect the kidney in a rat model of DN [39].

Cuproptosis genes were believed to be involved in nutrient metabolism, cell apoptosis and immune microenvironment regulation. In order to further elucidate the pathogenesis of DN and describe the heterogeneity, we explored the correlation between cuproptosis and immune microenvironment. Multiple cuproptosis genes expression were correlated with immune cell subtypes infiltration and with immune scores. Compared with normal samples, FDX1, LIAS, DLAT and DBT were lower expressed in DN samples. In DN samples, these four genes showed a trend of positive correlation, and a trend of negative correlation with most immune cells infiltration. GLS were highly expressed in DN tissues. However, there was no correlation between GLS expression and the above four genes in DN samples. GLS was also negatively correlated some kinds of immune cell subtypes infiltration. In order to describe the heterogeneity of the disease through typing based on immune microenvironment, we conducted consensus clustering via the expression levels of FDX1, LIAS, DLAT, DBT and GLS. We divided the samples into cluster C1 and cluster C2. Cluster C1 was characterized by low cuproptosis genes expression and high immune cell subtypes infiltration, while cluster C2 was the opposite. PCA based on global transcriptome gene expression showed that samples of different clusters enriched in different regions in the two-dimensional plane. This indicated that in the clustering model proposed by us, there was a good degree of differentiation between different cluster, which can be applied to describe the essential characteristics of the disease. The GSVA for cluster C1 and C2 also confirmed this view. Immune-related pathways were significantly enriched in cluster C1 and metabolism-related pathways were significantly enriched in cluster C2. This suggested that the samples in cluster C1 and C2 involved different mechanism pathways and were fundamentally different. In our study, due to the lack of clinical information, we were not able to explore the relationship between typing and clinical characteristics, which is a great pity. We hoped to clarify the relationship between immune microenvironment and patient prognosis in future study.

Unprecedented development of bioinformatics and machine learning algorithms, machine learning technology, especially deep learning, is widely used in various bioinformatics fields, and researchers in this field have proposed many important computational models, for example, GCNCRF algorithm predicts human lncRNA-miRNA interaction, Network distance analysis model lncRNA-miRNA association prediction (NDALMA) and deep learning prediction model DMFGAM were used to predict Human ether A-Go-Go-related gene (hERG) blockers, providing a new method for screening disease biomarkers [40–42]. To further search for new biological markers and quantify the risk of disease, we screened key genes via WGCNA related to clinical phenotypes and immune microenvironment characteristics. By combining the DEGs, we finally obtained two key genes, namely HRSP12 and DCXR. We used these two genes to construct an nomogram model.

We confirmed that the model had good diagnostic efficacy for DN. HRSP12 and DCXR were highly expressed in normal samples and in cluster C2 samples. HRSP12 and DCXR were negatively correlated with most immune cell subtypes infiltration. HRSP12 and DCXR may suppress the immune response. No previous studies have involved the relationship between these two genes and DN. Several studies elucidated the biological processes involved in these two genes. DCXR is involved in glycometabolism, with the ability to convert L-xylulose into xylitol, and the effect of detoxification. The deficiency of DCXR may be associated with a DM and cancer [43]. DCXR was a renal protective factor in chronic kidney disease. DCXR expression was down-regulated in glomerular tissue of chronic kidney disease, which was negatively correlated with disease severity and lead to poor outcome [44]. DCXR, which can also be called UK-114, is an RNA enzyme whose primary biological function is to degrade methylated RNA. Methylation is one of the ways in which epigenetic changes are thought to promote inflammatory responses in DN. Demethylation agents can prevent the progression of chronic kidney disease [45,46]. However, no studies have demonstrated the protective effect of DCXR on kidney. One previous confirmed that DCXR can promote Th2 cell differentiation, inhibit the release of inflammatory factors, and delay the progression of arthritis [47]. Another study indicated that DCXR was specifically expressed in liver and kidney tissue and was a potential biological marker of liver cancer [48].

The study of gene/protein signaling networks using ODE-based theoretical models has played a crucial role in disease prediction [49–52]. There are also some computational models based on deep learning that have high value for the in-depth study of diseases [53,54]. For example, a non-negative matrix decomposition deep learning model named MDA-AENMF is used to predict potential links between metabolites and disease [55], and researchers have proposed a new RNA-SEQ analysis framework called scAAGA that adaptively learns important genetic signatures from RNA-SEQ data to improve the clustering effect [56–59]. Compared with previous studies, our study has some shortcomings. First, our current study is based on a comprehensive bioinformatics analysis and cytological trials that require additional clinical or experimental evaluation of DN tissues to validate DCXR and HRSP12 expression levels. In addition, more detailed clinical data is needed to confirm the performance of predictive models. Finally, more detailed experiments are needed to elucidate the underlying mechanisms of DCXR and HRSP12 with DN.

## Conclusion

The pathogenesis of diabetic nephropathy (DN) is closely associated with immune microenvironment dysregulation and metabolic dysfunction. Cuproptosis genes, which play a regulatory role in both immune and metabolic processes, enable molecular stratification of disease subtypes, revealing heterogeneity in immune microenvironment profiles. Notably, HRSP12 and DCXR were identified as key regulators linking disease phenotypes and immune microenvironment features. These genes were integrated into diagnostic nomograms, demonstrating high accuracy and reliability for DN prediction. Furthermore, HRSP12 and DCXR were significantly upregulated in DN samples and exhibited negative correlations with immune cell infiltration and immune scores, suggesting their potential as diagnostic biomarkers and nephroprotective factors.

## Supporting information

**S1 Table.  Differentially expressed genes between control samples and diabetic nephropathy (DN) samples.**
(XLSX)

**S2 Table.  GO enrichment analysis of up-regulated genes in DN samples.**
(XLSX)

**S3 Table.  KEGG enrichment analysis of up-regulated genes in DN samples.**
(XLSX)

**S4 Table.  GO enrichment analysis of down-regulated genes in DN samples.**
(XLSX)

**S5 Table. KEGG enrichment analysis of down-regulated genes in DN samples.**
(XLSX)

**S6 Table. Gene set variation analysis of cluster C1 and cluster C2.**
(XLSX)

**S7 Table. The top 2 important feature variables of each model.**
(XLSX)

## Author contributions

**Conceptualization:** Fan Lu.

**Data curation:** Liping Guo, Yingying Yuan.

**Formal analysis:** Yuxuan Cao, Yingying Yuan.

**Methodology:** Yuxuan Cao, Liping Guo.

**Resources:** Hui Li.

**Software:** Hui Li.

**Writing – original draft:** Fan Lu.

**Writing – review & editing:** Hongmin Luo, Fan Lu.

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
