## [Decision Letter · Decision Letter 0]

21 Jun 2024

PONE-D-24-19973

Cuproptosis gene characterizes the immune microenvironment of diabetic nephropathy.

PLOS ONE

Dear Dr.  lu,

Thank you for submitting your manuscript to PLOS ONE. After careful consideration, we feel that it has merit but does not fully meet PLOS ONE’s publication criteria as it currently stands. Therefore, we invite you to submit a revised version of the manuscript that addresses the points raised during the review process.

Comments from PLOS Editorial Office: We note that one or more reviewers has recommended that you cite specific previously published works. As always, we recommend that you please review and evaluate the requested works to determine whether they are relevant and should be cited. It is not a requirement to cite these works. We appreciate your attention to this request.

We look forward to receiving your revised manuscript.

Kind regards,

Qi Zhao

Academic Editor

PLOS ONE

Journal Requirements:

   "This work were supported by Hebei Medical Science Research Key Project, No: 20221236. "

4. In this instance it seems there may be acceptable restrictions in place that prevent the public sharing of your minimal data. However, in line with our goal of ensuring long-term data availability to all interested researchers, PLOS’ Data Policy states that authors cannot be the sole named individuals responsible for ensuring data access (http://journals.plos.org/plosone/s/data-availability#loc-acceptable-data-sharing-methods).

Reviewers' comments:

Reviewer's Responses to Questions

**Comments to the Author**

1. Is the manuscript technically sound, and do the data support the conclusions?

Reviewer #1: Yes

Reviewer #2: Yes

2. Has the statistical analysis been performed appropriately and rigorously? 

Reviewer #1: Yes

Reviewer #2: Yes

3. Have the authors made all data underlying the findings in their manuscript fully available?

Reviewer #1: Yes

Reviewer #2: Yes

4. Is the manuscript presented in an intelligible fashion and written in standard English?

Reviewer #1: No

Reviewer #2: Yes

5. Review Comments to the Author

Reviewer #1: 1.English expressions need to be edited more careful and more native, in this manuscript, there are some mistakes.

2. I suggest the authors should add a flowchart in the manuscript to show the process very well.

3. I suggest the authors should elaborate their motivation in the discussion section, as they use several common bioinformatics analysis methods and online tools. What is the novelty and technicality of their work?

4. Advancements in computational capabilities have led to the widespread application of machine learning techniques, especially deep learning, across various bioinformatics fields. Important computational models in these fields should be cited. Some recommended studies are helpful (PMIDs: 36305458, 34232474, 35817399, 36584603, 37525507, 37660567, 38040204, and 37466194).

5. The authors should carefully check and unify the information of references. Some references lack the information of volume or contain the wrong page number.

6. Methods section is relatively simple because of no detail about the analysis. The analysis methods and statistical parameters must be clearly emphasized (i.e, reasons for selecting the algorithms used, threshold values used in statistical analysis, etc.).

7. The authors claim that HRSP12 and DCXR may be potential biological markers and renal protective factors. I think more validation should be adopted. Especially, I want to see some associations can be confirmed by other method or biological experiments.

8. Future work and limitations of the proposed algorithm should be addressed detailly in the manuscript for further research.

Reviewer #2: This study investigates the relationship between cuproptosis genes and the immune microenvironment in diabetic nephropathy (DN). RNA sequencing data from DN and normal renal tissues were analyzed for differential gene expression and immune cell subtype infiltration. NLRP3 and CDKN2A were positively correlated with immune scores, while NFE2L2 and other genes were negatively correlated. Consensus clustering based on cuproptosis genes identified two clusters: C1 (low cuproptosis gene expression, high immune cell infiltration) and C2 (opposite characteristics). Key genes DCXR and HRSP12 were identified and used to construct a nomogram with high diagnostic accuracy for DN. These genes may serve as potential biomarkers and renal protective factors. There are some issues that need to be addressed before the article can be accepted.

1.What do “NLRP3, CDKN2A, NFE2L2, LIAS, LIPT1, DLD, DBT and DLST” represent in the Abstract section? What are their full names? When first mentioned, the author should provide their full names to help readers better understand.

2.The authors should add a flowchart in the manuscript to show the process very well.

3.The Methods section in the text lacks corresponding literature references. The author did not provide sources and citations for the Methods mentioned.

4.The labeling of Figures in the paper is quite small. The contents within the figures are unclear. The author needs to carefully revise and modify the figure.

5.The authors should carefully check and unify the information of references. Some references lack the information of page number, such as ref [26].

6.The critique of current research limitations is somewhat lacking. The authors are encouraged to frame these deficiencies as directions for future research. Studies on gene/protein signaling networks using ODE-based theoretical modeling have played a crucial role in disease prediction (https://doi.org/10.1016/j.chaos.2023.114328,
https://doi.org/10.1103/PhysRevE.108.064412, PMIDs: 33389663, 35958114). It would be beneficial if the authors could integrate and reference these four studies in this paper.

7.Besides, deep learning based algorithms offer valuable insights into diseases. It is essential to discuss and cite key computational models in these areas. Including some recommended studies would be beneficial (PMID: 38737196, doi:10.1016/j.swevo.2024.101567).

6. PLOS authors have the option to publish the peer review history of their article (what does this mean? ). If published, this will include your full peer review and any attached files.

**Do you want your identity to be public for this peer review?** For information about this choice, including consent withdrawal, please see our Privacy Policy .

Reviewer #1: No

Reviewer #2: No

---

## [Author Response · Author response to Decision Letter 1]

13 Oct 2024

Thank you for your work. We have made changes in the manuscript. All the research data used were uploaded as a supplementary information package.

---

## [Decision Letter · Decision Letter 1]

17 Dec 2024

PONE-D-24-19973R1Cuproptosis gene characterizes the immune microenvironment of diabetic nephropathy.PLOS ONE

Dear Dr. lu,

Thank you for submitting your manuscript to PLOS ONE. After careful consideration, we feel that it has merit but does not fully meet PLOS ONE’s publication criteria as it currently stands. Therefore, we invite you to submit a revised version of the manuscript that addresses the points raised during the review process.

We look forward to receiving your revised manuscript.

Kind regards,

Vinay Kumar, Ph.D.

Academic Editor

PLOS ONE

**Additional Editor Comments:**

All comments from reviewer 1 needed to be answered

Reviewers' comments:

Reviewer's Responses to Questions

**Comments to the Author**

1. If the authors have adequately addressed your comments raised in a previous round of review and you feel that this manuscript is now acceptable for publication, you may indicate that here to bypass the “Comments to the Author” section, enter your conflict of interest statement in the “Confidential to Editor” section, and submit your "Accept" recommendation.

Reviewer #1: (No Response)

Reviewer #2: All comments have been addressed

2. Is the manuscript technically sound, and do the data support the conclusions?

Reviewer #1: Yes

Reviewer #2: Yes

3. Has the statistical analysis been performed appropriately and rigorously? 

Reviewer #1: Yes

Reviewer #2: Yes

4. Have the authors made all data underlying the findings in their manuscript fully available?

Reviewer #1: Yes

Reviewer #2: Yes

5. Is the manuscript presented in an intelligible fashion and written in standard English?

Reviewer #1: No

Reviewer #2: Yes

6. Review Comments to the Author

**Reviewer #1: ** 1. The authors didn't know how to write response letter. I always need to search for the revised content in the manuscript. You should list all the corresponding revisions in the letter.

2. The authors even write two “3” in the response to Reviewer #1.

3. The authors fail to elaborate their motivation in the discussion section. There are many papers on using machine learning to screen the core genes, so there is not as main innovation of the paper.

4. The authors should revise carefully corresponding to my previous comment #4. All the related paper should be cited.

5.The flowchart in Fig.1 is too simple to show the process of data collection and method implementation very well, please improve it.

6. Why the authors mention that HRSP12 and DCXR may be potential biomarkers and renal protective factors to my previous comment #7. If the authors can't get positive results, then what's the point of the experiment.

7. There are still some grammatical errors in the article, and the expression of some of the content is not clear enough. The authors need to check the manuscript carefully and make corresponding revision.

8. Why there are two new co-first authors in the revised manuscript?

**Reviewer #2: ** The author has addressed the reviewers' questions well. I recommend that the article be accepted for publication in this journal.

7. PLOS authors have the option to publish the peer review history of their article (what does this mean? ). If published, this will include your full peer review and any attached files.

**Do you want your identity to be public for this peer review?** For information about this choice, including consent withdrawal, please see our Privacy Policy .

Reviewer #1: No

Reviewer #2: No

---

## [Author Response · Author response to Decision Letter 2]

31 Jan 2025

Dear reviewer, we have modified the article according to your suggestion, thanks for your hard work.

---

## [Decision Letter · Decision Letter 2]

11 Mar 2025

Cuproptosis gene characterizes the immune microenvironment of diabetic nephropathy.

PONE-D-24-19973R2

Dear Dr. lu,

We’re pleased to inform you that your manuscript has been judged scientifically suitable for publication and will be formally accepted for publication once it meets all outstanding technical requirements.

Kind regards,

Vinay Kumar, Ph.D.

Academic Editor

PLOS ONE

Additional Editor Comments (optional):

Reviewers' comments:

Reviewer's Responses to Questions

**Comments to the Author**

1. If the authors have adequately addressed your comments raised in a previous round of review and you feel that this manuscript is now acceptable for publication, you may indicate that here to bypass the “Comments to the Author” section, enter your conflict of interest statement in the “Confidential to Editor” section, and submit your "Accept" recommendation.

Reviewer #1: All comments have been addressed

2. Is the manuscript technically sound, and do the data support the conclusions?

Reviewer #1: Yes

3. Has the statistical analysis been performed appropriately and rigorously? 

Reviewer #1: Yes

4. Have the authors made all data underlying the findings in their manuscript fully available?

Reviewer #1: Yes

5. Is the manuscript presented in an intelligible fashion and written in standard English?

Reviewer #1: Yes

6. Review Comments to the Author

Reviewer #1: The author has already addressed all of the issues. I think the manuscript can be published in PLOS ONE.

7. PLOS authors have the option to publish the peer review history of their article (what does this mean? ). If published, this will include your full peer review and any attached files.

**Do you want your identity to be public for this peer review?** For information about this choice, including consent withdrawal, please see our Privacy Policy .

Reviewer #1: No

---

## [Editor Report · Acceptance letter]

PONE-D-24-19973R2

PLOS ONE

Dear Dr. Lu,

I'm pleased to inform you that your manuscript has been deemed suitable for publication in PLOS ONE. Congratulations! Your manuscript is now being handed over to our production team.

Kind regards,

on behalf of

Dr. Vinay Kumar

Academic Editor

PLOS ONE